# Core Strategies to Increase the Uptake and Use of Potassium-Enriched Low-Sodium Salt

**DOI:** 10.3390/nu13093203

**Published:** 2021-09-15

**Authors:** Adefunke Ajenikoko, Nicole Ide, Roopa Shivashankar, Zeng Ge, Matti Marklund, Cheryl Anderson, Amy Atun, Alexander Thomson, Megan E. Henry, Laura K. Cobb

**Affiliations:** 1Resolve to Save Lives, Vital Strategies, New York, NY 10005, USA; funkeajenikoko@gmail.com (A.A.); nide@resolvetosavelives.org (N.I.); Shivashankar.r@icmr.gov.in (R.S.); Zge@resolvetosavelives.org (Z.G.); athomson.intern@resolvetosavelives.org (A.T.); 2Indian Council of Medical Research (ICMR), New Delhi 110029, India; 3Department of Epidemiology, John Hopkins Bloomberg School of Public Health, Baltimore, MD 21087, USA; mmarklund@georgeinstitute.org.au (M.M.); mhenry16@jhu.edu (M.E.H.); 4The George Institute for Global Health, University of New South Wales, Sydney 2042, Australia; 5Herbert Wertheim School of Public Health and Human Longevity Science, University of California, La Jolla, CA 92093, USA; c1anderson@ucsd.edu (C.A.); aatun@ucsd.edu (A.A.)

**Keywords:** low-sodium salt, sodium, sodium reduction, hypertension, cardiovascular disease

## Abstract

Excess sodium consumption and insufficient potassium intake contribute to high blood pressure and thus increase the risk of heart disease and stroke. In low-sodium salt, a portion of the sodium in salt (the amount varies, typically ranging from 10 to 50%) is replaced with minerals such as potassium chloride. Low-sodium salt may be an effective, scalable, and sustainable approach to reduce sodium and therefore reduce blood pressure and cardiovascular disease at the population level. Low-sodium salt programs have not been widely scaled up, although they have the potential to both reduce dietary sodium intake and increase dietary potassium intake. This article proposes a framework for a successful scale-up of low-sodium salt use in the home through four core strategies: availability, awareness and promotion, affordability, and advocacy. This framework identifies challenges and potential solutions within the core strategies to begin to understand the pathway to successful program implementation and evaluation of low-sodium salt use.

## 1. Background

Cardiovascular disease (CVD) is the leading cause of death globally, accounting for over 17 million deaths per year, of which 82% are in low- and middle-income countries (LMICs) [1]. Hypertension is a leading risk factor for CVD and death worldwide [2]. Excess sodium consumption and insufficient potassium intake contribute to high blood pressure and thus increase the risk of heart disease and stroke [3,4,5]. An estimated 1.7 million CVD deaths per year can be attributed to a high sodium intake [6].

Globally, the average salt intake is nearly double the maximum limit of 5 g per day set by the World Health Organization (WHO) [7,8]. In 181 of 187 countries (99% of the world’s population), the estimated average levels of salt intake exceed this limit [9]. At the same time, potassium consumption, which can reduce blood pressure and the risk of cardiovascular disease and stroke in adults, is below the WHO recommendation of 3510 mg per day in many countries [10,11]. Diets high in processed foods and low in fresh fruit and vegetables often lack potassium.

Reducing sodium and increasing potassium intake can significantly reduce blood pressure in adults [12,13]. A sodium-to-potassium ratio (sodium intake divided by potassium intake) of ≤1 is associated with a reduction in the risk of cardiovascular events [14,15]. A systematic review suggested that this ratio is more strongly associated with lowering blood pressure than the levels of either sodium or potassium intake alone in hypertensive populations [13].

In low-sodium salt, a portion of sodium chloride (typically ranging from 10 to 50%) is replaced with alternative minerals such as potassium chloride (KCl), calcium chloride (CaCl_2_), magnesium chloride (MgCl_2_), or magnesium sulphate (MgSO_4_). Low-sodium salts most commonly reduce sodium content through substitution with KCl. For the purpose of this paper, potassium-enriched low-sodium salt will be referred to as low-sodium salt, and sodium chloride (NaCl) will be referred to as regular salt. Randomized trials [16,17,18,19] have shown that low-sodium salt is effective in reducing both systolic and diastolic blood pressure, and modeling as well as a large-scale trial have documented reductions in cardiovascular disease and mortality [20,21]. This impact is likely not due to sodium reduction alone; a recent trial found a reduction in blood pressure despite urinary excretion data showing an increase in potassium intake but no change in sodium intake [19]. Animal studies suggest increased potassium intake may increase sodium excretion [22,23], potentially explaining why a reduction in urinary sodium excretion is not observed in low-sodium salt trials. One study found that potassium acts like a thiazide diuretic, a major class of blood pressure medication, on NaCl cotransporters (NCC) which serve to move NaCl through the body. Potassium reduced the presence of NCC, which led to a reduction in urinary sodium excretion [23]. Like regular salt, low-sodium salt can be fortified with iodine for enhanced synergy between two important global public health nutrition programs.

Dietary sodium comes from three major sources: (1) salt added during cooking or at the table (discretionary salt), (2) from packaged food, and (3) from food prepared outside the home (e.g., restaurant and street food). In many LMICs, discretionary salt has been reported to be the largest source of sodium [24], but packaged and out-of-home food sales are increasing in these regions, often at a faster rate than in high-income countries [25,26,27]. Traditional strategies to reduce discretionary salt have relied on behavior change communications. However, structural or policy interventions are more effective than those that rely solely on individual behavior change [28,29]. Policies that promote the use of low-sodium salt may be a more effective, scalable, and sustainable approach to reduce sodium from discretionary salt and can also be used to reduce sodium in packaged foods or food prepared outside the home such as in restaurants and canteens [30]. 

### 1.1. Country Examples

While several countries are currently implementing salt reduction programs [31], population-level low-sodium salt intervention studies have only been implemented in two countries (see Table 1). A trial in Peru promoting low-sodium salt through a social marketing campaign found that it was both acceptable to consumers and effective at reducing blood pressure and hypertension incidence [19]. In China, a trial found that, while promotion of low-sodium salt increased uptake, this was much more pronounced when a subsidy was added on top of the promotion to offset the higher cost of low-sodium salt [32]. Other salt reduction programs in China have also included low-sodium salt promotion. For example, the Beijing government promoted the uptake of low-sodium salt in 2010, strengthening the supply chain so that restaurants, institutional canteens/cafeterias, and families could purchase low-sodium salt from 28 supermarket chains [33]. Low-sodium products were also promoted in markets and restaurants and through public education activities as part of the Shandong-Ministry of Health Action on Salt and Hypertension (SMASH) program. At the end of the program, a nearly eight-fold increase in the sales of low-sodium salt was found, along with a 25% reduction in sodium intake and a 15% increase in potassium intake [34].

### 1.2. Challenges to Scaling up Low-Sodium Salt

Despite the benefits, low-sodium salt is not commonly used in most countries. Some of the major barriers to scaling up include availability, awareness, cost, taste, and safety concerns.

#### 1.2.1. Availability

Low-sodium salt is not widely available on the global market, particularly in LMICs [35]. A recent study found that low-sodium salt is available in just under a quarter of all countries [35]. Production and regulatory challenges (e.g., development of national standards on low-sodium salt, consistent definitions of acceptable levels of sodium and potassium, etc.) will have to be addressed to allow increased production of low-sodium salt. Low availability may also be due to low demand.

#### 1.2.2. Awareness

Many consumers are not aware of the existence of low-sodium salt and of its benefits. A study in Shenzhen, China, found that nearly half of the local population had never heard of low-sodium salt; others simply did not think of it when grocery shopping [36]. Health care providers may not be familiar with the product either. A study of health care providers in Mangalore, India, found that only 53% of the participants knew about the benefits of low-sodium salt, and only 19% had recommended it to patients [37]. Increasing awareness of low-sodium salt among health care providers and the general public could help generate consumer demand. Several countries display health warnings/advice on low-sodium salt targeting those following a low-potassium diet, with kidney disease, or using anti-hypertensive medications. However, there is no standardization of warning labels across countries or manufacturers, and these labels may leave consumers confused.

#### 1.2.3. Cost

Cost is also a barrier to low-sodium salt use, which is particularly concerning for individuals with a lower income level [36]. It costs 1.6 to 6.0 times more than regular salt, a reflection of the higher cost of potassium compared to sodium [35]. Regular salt, an essential commodity, is sold at subsidized rates in some countries, which further increases the cost difference between the two products [38]. Low-sodium salt is also marketed as a premium health product in some places. A trial in rural northern China found that participants in intervention sites who received subsidies plus education purchased nearly double the amount of low-sodium salt as those who received education only [32]. Even where awareness is high, cost-lowering measures may be necessary to increase low-sodium salt use.

#### 1.2.4. Taste

A recent review explored whether replacing regular salt with low-sodium salt in cooking alters its flavor and palatability [17]. Higher levels of KCl are associated with a metallic taste; a lower percentage of KCl combined with taste enhancers (e.g., food acids, amino acids, and umami substances) and/or the use of other processing techniques may improve the flavor. A recent review of the literature on low-sodium salt found that most individuals were unable to distinguish between food prepared with regular salt and that containing low-sodium salt when ≤30% of the sodium was replaced by KCl [17]. A trial in India found that participants rated the taste of low-sodium salt (70% NaCl, 30% KCl) well and they incorporated it into their daily cooking habits [39]. A major benefit of low-sodium salt is that it reduces sodium while maintaining the salty flavor of food; therefore, even individuals addicted to salty foods—due to high consumption of processed food or to adding salt at home—can still reduce their sodium intake while maintaining taste [40,41]. Low-sodium salt is a viable strategy to bring down overall sodium consumption levels and should be accompanied by longer-term efforts to comprehensively reduce the overall amount of salt in the broader food environment. While taste can be seen as a barrier to the adoption of low-sodium salt, maintaining ≤30% KCl in low-sodium salt minimizes issues with taste. Some food products may have a narrower level of acceptability of potassium levels than others.

#### 1.2.5. Safety

Concern regarding hyperkalemia (high levels of potassium in blood) is one of the main challenges to scaling up low-sodium salt use. Low-sodium salt is safe for most people. However, there are concerns that the additional potassium intake resulting from low-sodium salt use may not be well tolerated in certain subpopulations and could increase the risk of hyperkalemia and its adverse effects, including arrhythmias and sudden cardiac death. Individuals who are most vulnerable to hyperkalemia are those with advanced chronic kidney disease (CKD); other vulnerable individuals are those who are on medications that impair potassium excretion (e.g., potassium-sparing diuretics or medications that affect the angiotensin system). There is insufficient evidence regarding the effects of low-sodium salt on the occurrence of hyperkalemia, largely due to ethical issues related to enrolling participants with kidney disease in trials in which they would be exposed to potassium [17]. However, many studies excluded participants based on a self-reported history of chronic kidney disease [19,21,39] and therefore may have included individuals with advanced CKD who were unaware of their disease status. A large-scale trial on low-sodium salt that followed 21,000 high-risk adults in China over five years reported no increased risk of serious adverse events attributable to low-sodium salt use [21]. The trial excluded patients using potassium-sparing diuretics or potassium supplements and patients with serious kidney disease. Despite the uncertainty around hyperkalemia, there is a large net benefit of using low-sodium salt at the population level; a recent modeling exercise demonstrated that even within the sub-population of those with chronic kidney disease, there would be net lives saved [20]. Addressing safety concerns will require an increased understanding of the impact on potentially vulnerable populations, coupled with strategies to identify those at risk and ensure that the risk is communicated appropriately [42].

## 2. Developing a Framework

In this paper, we propose a framework to address the major challenges to increasing the uptake and use of low-sodium salt. The framework (Table 2) specifies four core strategies and their related challenges: availability, awareness and promotion, affordability, and advocacy. Although taste is listed above as a potential challenge, it should not impede implementation if the salt is ≤30% KCl. The challenges, along with potential solutions, are explored in the sections below. The potential solutions draw on existing literature and case studies from other areas of public health that can be applied to low-sodium salt.

## 3. Applying Framework: Strategies for Scaling up the Use of Low-Sodium Salt

### 3.1. Availability in the Market

#### 3.1.1. Addressing Production or Regulatory Challenges

Producers and distributors of low-sodium salt will benefit from a level playing field with clear rules, regulations, and enforcement. This includes defining acceptable levels of sodium, potassium, and other minerals, what the products can be called, and how to fortify the products with iodine. Regulations can define allowable health claims on packages and specify required warning statements regarding hyperkalemia which apply to all companies. For example, in 2015, China released food safety national standards for edible salt, which defined low-sodium salt by the use of food additives such as KCl (between 10–35%) to replace NaCl. They also required low-sodium salt to be iodized [43].

#### 3.1.2. Distribution/Supply Chain Issues

Incentivize/subsidize manufacturers to produce low-sodium salt: Governments have the potential to increase the production of low-sodium salt through incentives and subsidies directed at manufacturers. To our knowledge, no country currently offers incentives for manufacturers to produce low-sodium salt, although several countries have used incentives to increase the production of other products like iodized salt. In India, the government has subsidized potassium iodate to increase the production, demand, and supply of iodized salt, resulting in an eight-fold increase in the national production of iodized salt (from 0.7 million metric tons (MMT) in 1985–1986 to ~6.2 MMT in 2013) [44]. This suggests that governments can use subsidies to increase the production of low-sodium salt and possibly decrease production costs. 

Improve the supply chain: Governments can ensure that low-sodium salt is widely available for public purchase by strengthening the supply chain. In Beijing, the government strengthened the supply chain so that restaurants, institutional canteens, and families could purchase low-sodium salt from supermarket chains [33]. There are also alternative supply routes that can be explored for specific populations. In 2018, the government of the Indian states of Gujarat and Madhya Pradesh made double-fortified salt (salt fortified with iodine and iron) available through the public food distribution system. To help the government roll out the salt, Nutrition International, a global non-profit organizaiton, helped the state governments to establish quality control and quality assurance protocols, including training of government staff at various levels [45]. 

Increase demand for low-sodium salt: Without demand, manufacturers have little incentive to increase the production of low-sodium salt. Demand can be generated through subsidies (described in Section 3.3.1), consumer awareness, and promotion activities (further discussed in Section 3.2). In the SMASH program, the salt industry took advantage of the government’s salt reduction education intervention to invest in low-sodium salt and promote it widely, which led to an eight-fold increase in the sales [34]. Governments can also increase the demand by requiring that publicly funded institutions procure low-sodium salt rather than regular salt.

### 3.2. Awareness and Promotion—Generating Consumer Demand for Low-Sodium Salts

#### 3.2.1. Awareness

##### Consumer Education on the Benefits of Low-Sodium Salts for Health

Multiple strategies can be used to educate the consumers on the health benefits of low-sodium salt. Mass media campaigns can be used to motivate individuals to consume less salt by highlighting the dangers of a high-sodium diet and recommending low-sodium salt as a specific strategy to reduce sodium intake. While few studies have looked at awareness campaigns for low-sodium salt [32], several have shown the effectiveness of awareness campaigns in sodium reduction more broadly [46,47]. For example, a mass media campaign in South Africa was found to increase public awareness of the association between high salt intake, blood pressure, and CVD in the three South African provinces evaluated. Post-intervention, significantly more participants reported taking steps to control their salt intake (38.0% increased to 59.5%) [48]. 

Awareness campaigns are most effective when conducted as part of a multi-strategy program. In Vietnam, the Communication for Behavioral Impact (COMBI) framework was utilized to reduce population salt intake through mass media communication, interventions in primary schools and community communication programs, resulting in reductions in mean salt excretion and improved knowledge and behaviors following the intervention [49]. In India, public education and intensive social mobilization activities were conducted through various channels including print media, television, and radio to create consumer demand for iodized salt, with the national uptake of iodized salts reaching 51% in 2005–2006 and 71% in 2009 [44].

Increased awareness can lead to greater demand for low-sodium salt. Campaigns should be done in tandem with efforts to improve the availability and reduce the relative cost of low-sodium salt, as building awareness for something consumers cannot readily obtain will be counterproductive.

##### Healthcare Providers as a Vehicle for Consumer Education

Studies have shown that consumers who received advice from healthcare providers are more likely to change or attempt to change their behavior [50]. Many providers offer lifestyle advice to patients and could include the recommendation to use low-sodium salt, particularly for patients with hypertension or pre-hypertension. Providers should accompany recommendations to use low-sodium salt with messages to reduce overall sodium consumption. In some cases, healthcare providers may be able to prescribe low-sodium salt to the appropriate hypertensive patients at reduced or no cost to the patient. This would be similar to producing prescription programs for food-insecure patients with NCDs [51,52,53]. This approach has the added advantages of increasing potassium consumption by people at the highest risk and avoiding recommending it to patients with CKD and patients at risk of developing hyperkalemia.

In a salt reduction program in Fiji, healthcare providers were trained to educate consumers, and food businesses were engaged through targeted consumer behavior change programs to reduce salt in meals, which led to a 1.4 g/day reduction in sodium intake (11.7 g/day to 10.3 g/day; *p* = 0.115) [54]. Working with professional societies and incorporating low-sodium salt into existing healthcare provider education courses may be effective for scale-up.

#### 3.2.2. Promotion

##### Social Marketing

Social marketing is the application of commercial marketing principles (branding, product design, appropriate pricing, sales and distribution, and communications) to influence social behavior [55]. It seeks to change behavior through comprehensive, multifaceted approaches that provide coordinated interventions to specific audiences. A strong social marketing intervention will promote a product (e.g., low-sodium salt), while considering the price (cost, perception of safety, and access) and place (retail settings). Social marketing to promote low-sodium salt in Peru led to a reduction in blood pressure and an increase in potassium intake [19]. This approach has been shown to change many health-related behaviors and has the potential to influence consumers to use low-sodium salt and bring awareness of the dangers of a high-salt diet [56].

The 4 Ps of marketing (product, placement, price, and promotion) are marketing tools used by food retailers that can be incorporated into social marketing efforts as well. Food choices are influenced by access and availability in retail settings. Generally, supermarket layouts are designed to encourage the purchase of unhealthy products such as chips, soda, and candy that are prominently displayed at check-out counters and ends of aisles, discounted, and attached to in-store promotions [57,58]. Studies indicate that product promotion (e.g., food tastings, signage) and prominent placement are the most effective strategies for increasing sales of healthy food within retail settings [59]. A trial in the United States found that using product placement, signage, and increasing the number of products on display increased the sales of targeted healthy food items [60]. Similarly, another study found that placing non-alcoholic beverages in end-of-aisle displays increased their sales volume [61]. In-store promotion strategies may improve the sales of low-sodium salt.

Social marketing campaigns often take the place of advertising and other promotion activities done by the commercial sector. If companies are willing to invest in the promotion of low-sodium salt themselves, less investment in social marketing may be required.

### 3.3. Affordability

#### 3.3.1. Subsidies

Subsidies are particularly powerful tools to promote consumption and can be used to reduce or eliminate the retail price difference between low-sodium and regular salt. Government subsidies can be provided to producers during manufacturing (discussed in Section 3.1.2) or applied at other points during the distribution process, including at points of sale for consumers. 

Subsidies are often invisible to consumers but ultimately create fiscal incentives for consumers to purchase targeted foods—in this case, low-sodium salt. In Beijing, the government provided a subsidy by offering a bonus of 75 g of low-sodium salt and a salt restriction spoon (both paid for by the government) when consumers purchased 400 g of low-sodium salt [33]. Supplemental Nutrition Assistance Program (SNAP) recipients in the United States received subsidies (30% price reduction) on fruit and vegetable purchases applied at checkout through a pilot study, which resulted in increased consumption of fruit and vegetables [62]. Subsidizing products within social protection programs like SNAP may be helpful to reach consumers for whom price is a barrier, without requiring governments to subsidize across the board. Some countries have considered using taxation of regular salt or of unhealthy food products to pay for subsidies for low-sodium salt and other healthier alternatives.

#### 3.3.2. Vouchers

Government-funded voucher programs can increase access to targeted products, particularly among those for whom cost is a major barrier to a healthy diet. Vouchers are coupons provided to consumers to purchase specific products. Vouchers differ from subsidies because they act as cash in retail settings and are used at points of sale. Program costs to implement vouchers may prohibit this strategy over the long term, but vouchers can temporarily be used to build demand and awareness for low-sodium salt and incentivize consumers for whom price is a barrier. 

A trial in China found that participants in intervention villages who received vouchers plus education purchased nearly double the amount of low-sodium salt as did those who received education only [32]. Some lessons from fruit and vegetable and other e-voucher programs may also be applicable. Voucher programs have been used to increase fruit and vegetable intake and often are offered in conjunction with nutrition supplement programs in the United States. For example, in New York City (NYC), the Health Bucks program provides $2 USD coupons for every $2 USD spent (up to $10 USD per day) at NYC farmers markets to food assistance recipients to purchase fresh fruits and vegetables [63]. 

#### 3.3.3. Taxation of Regular Salt

Another potential method to boost low-sodium salt use is to tax regular salt. Salt taxes alone are unlikely to increase the purchase of low-sodium salt, but they can be used to raise revenue to subsidize low-sodium salt and to equalize the cost to consumers of regular and low-sodium salt. Additionally, salt taxes on regular salt and/or on other high-sodium foods can incentivize the industry to reformulate packaged food and market healthier products like low-sodium salt. Lessons from taxes on sugar-sweetened beverages show that taxes must be substantial to be effective [64]; in order to reduce the financial burden that taxing salt and high-sodium foods may add to lower-income populations, subsidies and education to consume less salt may help.

### 3.4. Advocacy

To be successful at a population level, programs to scale up low-sodium salt use will need to be led by governments. However, the initial push for these programs can also come from champions in different sectors, including the medical field, nutrition groups, and civil society.

#### 3.4.1. Addressing Concerns on Hyperkalemia

Concerns regarding hyperkalemia are one of the main challenges to implementing low-sodium salt programs. Working with key stakeholders to ensure that that the risks and benefits are clearly understood is crucial. It may be necessary to undertake a risk assessment. For example, the UK Scientific Advisory Committee on Nutrition conducted a risk assessment to determine if food manufacturers should be encouraged to use low-sodium salt to reduce salt in their products. They determined that the benefits outweighed the risk, clearing the path for the government to move forward [65].

In addition, there are measures that can be taken to minimize the risks of hyperkalemia. Many companies have addressed the potential for hyperkalemia by placing warning statements on low-sodium salt. These warning statements can be regulated and standardized by governments in order to inform consumers of possible adverse effects of increased potassium intake, which may increase the risk for hyperkalemia [66]. However, the language used in these warnings should not discourage consumers who are *not* at risk. The warning statements should (1) raise consumers’ awareness that the low-sodium salt contains potassium and (2) inform individuals who have been specifically instructed to limit their dietary potassium intake to consult their physician or other health professionals prior to use. Potential warning statements are outlined in Box 1. Other risk mitigation measures to consider, particularly if low-sodium salt will be used in the packaged and restaurant food industries, are increased screening for chronic kidney disease, ensuring that the potassium contents of packaged foods is on the label, and standardizing recipes in restaurants.

Box 1Potential warning statements for potassium-enriched low-sodium salt.
**
*Example of warning statement:*
**

*This product contains potassium, a necessary nutrient. If you have been told to limit potassium in your diet, please consult your doctor before use.*



While low-sodium salt is an innovative and potentially high-impact strategy to reduce sodium consumption, concerns about the adverse effects in individuals with hyperkalemia are legitimate. Therefore, it is still recommended that individuals with CKD or at risk of hyperkalemia consult their physician before consuming low-sodium salt.

#### 3.4.2. Securing Commitment from Key Stakeholders

##### Advocate for Political Buy-In

Effective low-sodium salt interventions at scale are government-supported. Although one agency, such as a food regulator or public health agency, might coordinate and lead efforts, there will likely be a need for collaboration across sectors and to engage a wide range of government stakeholders (e.g., food industry, trade, mining). Educating government stakeholders can be done either directly by the lead agency or through external partners such as civil society advocates and health experts. Different working groups can be engaged in leading the advocacy work to promote low-sodium salt. For example, civil society organizations, local champions, and medical associations can incorporate advocacy for low-sodium salt into their activities. Researchers can build evidence by modeling the impact on lives and the money saved by scaling use of low-sodium salt [20]. 

##### Advocate for Industry Support

The salt industry is potentially a critical partner in any effort to scale up low-sodium salt use, as it is responsible for manufacturing, distributing, and marketing the product. Engaging with the industry is a first step, and a successful program will need to identify and address the current barriers to increasing the production and distribution of low-sodium salt. When meeting with representatives from the industry, benefits to the industry should be clearly defined. In the SMASH project in Shandong, China, the salt industry (which was government-owned at the time) was a key partner. The program’s salt reduction education aligned with the industry’s objectives as it was able to realize a higher profit with low-sodium salt compared to regular salt. Low-sodium salt increased from <1% to >25% of household-size salt packets purchased in this setting where the industry was engaged and active in production and promotion [34]. 

##### Civil Society as Advocates

Civil society organizations are critical advocacy partners. They can speak directly to the public or through the media and can help shape the public opinion toward planned government actions. They can also push for government action in new areas and encourage the government to prioritize salt reduction through scaling up low-sodium salt. Support and advocacy from the civil society has been critical to progress in efforts such as tobacco control. In the Latin America and Caribbean region, the support of the civil society led to the implementation of 100% smoke-free policies in Colombia, Guatemala, Panama, and Uruguay [67]. Tobacco control illustrates the power of civil society action against a powerful industry. In contrast, the salt industry can be an ally rather than an antagonist. In the United States, the Center for Science in the Public Interest (CSPI), Nutek (a low-sodium salt company), and other advocates worked together, successfully, to urge the Food and Drug Administration (FDA) to amend the guidelines for potassium labeling to permit manufacturers to use the term “potassium salt” instead of “potassium chloride”, which would help reformulated food products be more acceptable to consumers [68]. 

## 4. Discussion

In this article, we outlined a framework to address the major challenges to a successful scale-up of low-sodium salt use by employing four core strategies: availability, awareness and promotion, affordability, and advocacy. The solutions proposed in this article can make low-sodium salt more available and accessible to consumers, increasing its uptake, reducing blood pressure, and reducing cardiovascular disease. Increasing the uptake of low-sodium salt is an innovative and potentially high-impact approach to reducing the high sodium intake. While there are legitimate concerns about the potential for hyperkalemia, replacing regular salt with a 30% KCl low-sodium salt would be the equivalent of consuming potassium from 1–2 additional bananas per day. It remains advisable for individuals at risk of hyperkalemia to consult their physicians before consuming low-sodium salt. Low-sodium salt should be a complementary strategy to a robust sodium reduction program rather than the sole strategy implemented. It is critical to downregulate consumers preferences for salty foods through the implementation of comprehensive sodium reduction programs.

### Areas for Further Research

Although evidence from China and Peru suggests that a successful scale-up of low-sodium salt is possible, there has not yet been a population-wide implementation of sustainable government programs to scale up low-sodium salt. Additional evidence focusing on the implementation and impact of these interventions to improve the use of low-sodium salt is needed. For example, testing solutions in real-world settings regarding supply chain challenges, increasing purchasing through marketing strategies at the point of sale, and addressing affordability through taxation of salt or subsidies for low-sodium salt would advance our understanding of the key elements in an effective, comprehensive low-sodium salt use strategy.

Potassium-enriched low-sodium salt may not be the only way to reduce the consumption of regular salt. Food manufacturers are researching ways to alter the shape or crystal structure of salt to get more flavor with less NaCl. While these approaches have potential, particularly for packaged foods, they may not be equally useful for home-cooked foods and do not have the added health benefit of potassium. 

## 5. Conclusions

The promotion of low-sodium salt is a promising strategy to address excessive sodium and inadequate potassium consumption. This paper proposes an implementation framework to address the major barriers to the uptake and use of low-sodium salt, with examples on how these barriers can be overcome. The strategies outlined can inform efforts to increase the uptake of low-sodium salt as part of government-led sodium reduction programs in countries around the world.

## Figures and Tables

**Table 1 nutrients-13-03203-t001:** Population-level low-sodium salt intervention studies.

Peru [19]	China [32]
**Start date:** April 2014	**Start date:** May 2011
**Target**: Peruvian households	**Target**: Villages from five provinces in northern China
**Intervention components:** Replace regular salt in community households with low-sodium salt (75% NaCl and 25% KCl) Developed brand identity Social marketing campaign	**Intervention components:** Community-based education through public lectures, display and distribution of promotional materials, and special interactive education sessions Low-sodium salt (65% NaCl, 25% KCl, and 10% MgSO_4_) Price subsidies in 50% of intervention villages
**Results:** Reduction in systolic blood pressure (BP) of 1.92 mmHg (0.54, 3.29; *p* = 0.006) and in diastolic BP of 1.18 mmHg (0.08, 2.29; *p* = 0.036) among hypertensive patients Participants were 51% less likely (hazard ratio = 0.49, (0.34, 0.71), *p* < 0.001) to develop hypertension in the intervention period than in the control period Increase in mean potassium intake (mean difference of 0.63 g/day (0.47,0.78)) but no change in sodium intake	**Results:** Mean monthly sales of low-sodium salt per shop were: ○ 2 kg/month in the control group ○ 19 kg/month in the education group ○ 38 kg/month for the price subsidies + education group Reduction in mean urinary sodium excretion by 5.5% (1%, 26%; *p* = 0.03) in intervention villages Increase in potassium excretion by 16% (4%, 10%) and sodium to potassium ratio decrease by 15% (0.5%, 1.2%; *p* < 0.001)

**Table 2 nutrients-13-03203-t002:** Framework for addressing challenges to scale up low-sodium salt use.

Core Strategies	Challenges to Address
Availability	Lack of availability in global market
	Production and regulatory challenges
	Distribution and supply chain issues
Awareness and Promotion	Lack of awareness and demand from consumers and healthcare providers
Affordability	Higher cost of low-sodium salt
Advocacy	Concerns with hyperkalemia
	Lack of political will to invest sufficiently to ensure high uptake and widespread use of low-sodium salt

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
