# Peer review of "Core Strategies to Increase the Uptake and Use of Potassium-Enriched Low-Sodium Salt"

_nutrients, 2021, doi:10.3390/nu13093203_

Round 1

Reviewer 1 Report

The Authors of this paper argued on a very important issue. The Core Strategies to Increase the Uptake and Use of Potassium-Enriched Low-Sodium Salt were clearly described.
However, although several studies showed the safety and the benefit of the Potassium-Enriched Low-Sodium Salt consumption, the potential adverse effects due to hyperkalemia on some categories cannot be excluded. This is the first limitation of the use. Therefore, this topic should be better reported in the conclusion.

3.2.1.2, “Many providers give lifestyle advice to patients and could incorporate the recommendation to use low-sodium salt, particularly for patients with hypertension or pre-hypertension”. Patients with hypertension may have kidney disease or use RAAS inhibitors, hence at risk of hyperkalemia. They should discuss this limitation. 

Moreover, another limitation may be that the amount of discretionary salt intake does not change. They should also argue better on this topic. 

They should discuss on different types of low-sodium salt, and that the low-sodium salt used in China also included Magnesium.

  Minor points:
-Check references 5, 7, and 9
-Check line 411
-In addition to the references 3 and 4, they could consider a more recent reference (e.g. PMID: 29482962)

Reviewer 2 Report

The authors exposed in their review challenges and strategies to increase the use of potassium-enriched low-sodium salt. The topic is of great interest as a the results of SSaSS trial showing the beneficial effect of salt substitution on stroke, major cardiovascular events and death reduction have just been published in the latest NEJM issue online (B Neal et al: Effect of salt substitution on cardiovascular events and death). 

The review is well written and some of the authors have participated to previous papers analysing strategies do increase low-salt use in Low or Middle Income countries. 

Some interesting references could be added such as the NEJM trial previously mentioned. 

I would just suggest to add some precisions and references to the review.

Background: 

L46: ref 13 mainly shows the reduction in stroke. The paragraph on sodium/potassium ratio could be extended to show that the ratio probably better captures the effects of both electrolytes. Cut-off for the ratio to propose?

The effect of Potassium on salt excretion (L-58-59): there is a nice study showing how potassium acts on NCC having a thiazide-like effect. The mechanism should be mentioned (Sorensen MV. Kidney International 2013 - PMID 23447069)

1.2 Country Exemples: Other papers  mentioned other LIC or MIC countries:  Mongolia, Argentina, South Africa or Vietnam (e.g J Webster el al in Public Health Nutr 2021), this could be reported as well in the review. Also mentioned 

L79: sentence is confusing

1.3 challenges: I would withte awareness first (1.3.1) then availability and costs

In this part I would also add the "addictive" or dependency of salt, the importance in processed foods and bread in those having dependency or low income in high income countries (REF PLOS one 2015 -PMID 25692302; PMID 33934668).

1.3.3 Mentioned also in the cost that a higher diet in potassium implicates fruits and vegetables which can be problematic for those with lower incomes. Costs is probably the main concern or adherence either in LIC or even in HIC for people with lower social and education level. 

The education is also something important to underline: with education comes awareness and adherence...

1.35. Safety: The recent NEJM article did not describe differences in hyperkaliemia, but that is thrue that participants with known kidney disease were excluded. 

Minor. 

Ref 1 and 5 seem to be incomplete or delete accessed on
